# Study into the Evolution of Spatiotemporal Characteristics and Driving Mechanisms of Production–Living–Ecological Spaces on the Indochina Peninsula

Shuang Lu [1,†], Zibo Zhou [2,†], Mingyang Houding [1], Liu Yang [1], Qiang Gao [3,4], Chenglong Cao [1,*], Xiang Li [1,*] and Ziqiang Bu [3,4]

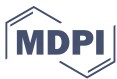

[1] College of Geoscience and Surveying Engineering, China University of Mining & Technology-Beijing, Beijing 100083, China; sqt2100203051@student.cumtb.edu.cn (S.L.); 2010290312@student.cumtb.edu.cn (M.H.); 108889@cumtb.edu.cn (L.Y.)

[2] Chinese Institute of Coal Science, Beijing 100013, China; zhouzibo@cumtb.edu.cn

[3] Institute of Geographic Sciences and Natural Resources Research, CAS, Beijing 100101, China; gaoqiang0155@igsnrr.ac.cn (Q.G.); buziqiang3314@igsnrr.ac.cn (Z.B.)

[4] College of Resources and Environment, University of Chinese Academy of Sciences, Beijing 100049, China

[*] Correspondence: ccl@student.cumtb.edu.cn (C.C.); lx@student.cumtb.edu.cn (X.L.); Tel.: +86-18811667538 (C.C.)

[†] These authors contributed equally to this work.

**Abstract:** Influenced by historical background, regional economic development, and the frequent occurrence of armed conflict, the human–earth relationship in the Central and Southern Peninsula, which is located in a "fragmented zone", is characteristic of the region. The Indochina Peninsula has now become an area of interest for the study of spatial changes in production–living–ecological spaces (PLES). Taking the Indochina Peninsula as the study area, this paper explores the evolution of the spatiotemporal patterns of PLES and its driving mechanism in the Indochina Peninsula, from 2010 to 2020, based on a grid scale. Methods such as the land-use transition matrix, land-use dynamics index, and geographically and temporally weighted regression (GTWR) were used in our model, which will provide the basic data and reference for sustainable development planning across the Indochina Peninsula. Our results show that, from 2010 to 2020, ecological space dominated the PLES pattern on the Indochina Peninsula, but its area gradually decreased, accompanied by a sharp increase in the areas of productive and living spaces. The area of PLES interconversion on the Indochina Peninsula in 2010–2020 was 212,818.70 km², and the intertransfer of production and ecological spaces was distributed in a networklike manner throughout the Indochina Peninsula, while the transfer of living space was distributed in a pointlike manner. The migration path of the center of gravity of PLES on the Indochina Peninsula demonstrated a significant directional difference, and the direction and extent of the standard deviation ellipse distribution of the ecological space was similar to that of the production space. The PLES's pattern evolution was affected by the degree of multiple factors, with a significant spatial and temporal heterogeneity. The positive and negative feedback effects of the factors were distributed in different areas and in different transfer directions.

**Keywords:** Indochina Peninsula; production–living–ecological spaces; GTWR

## 1. Introduction

Since the 1990s, there have been several mechanisms for geopolitical and economic co-operation across the Indochina Peninsula; such co-operation exists between countries as a top-level strategy and influences land-use changes on the Indochina Peninsula in different forms and dimensions, and to different degrees [1]. As the trend in regional political and economic integration intensifies and international attention to the Indochina Peninsula continues to grow, transregional economic cooperation and projects such as "alternative planting" and the construction of border roads have had a profound impact on land use

and land cover in the area [2–4]. Due to its special location and economic development mode, the Indochina Peninsula is now an active zone of land-use change [2]. Strengthening the study of land-use change across the Indochina Peninsula is of great significance for the improvement of its sustainable development, and for the ecological and environmental benefits of the region.

Research on land-use and land-cover change (LUCC) in the Indochina Peninsula has always been a hot academic topic, and previous studies have mainly focused on LUCC simulation and prediction using different remote sensing data products and improving the simulation accuracy to accurately assess the historical dynamics and future changes of LUCC in the region [5–8]. Indochina Peninsula countries have inconsistent LUCC driving factors due to differences in their social backgrounds, land systems, and topography, but due to geographic proximity, there are some similarities in the driving factors, and results of studies show that the LUCC in the Indochina Peninsula is mainly affected by the land policy and the economic market [8,9]. Due to the specificity of the driving mechanism of land-use change in border areas, some scholars selected the Indochina Peninsula borderline to carry out LUCC research and found that the border land-use change was affected by the local economy, policy, humanities, nature and the socio-economic development of the bordering countries [2,10].

The most important feature of regional land-use change is the mutual transformation between land-use types and their dominant functions, i.e., the mutual transfer between three land-use function types, namely, production space, living space, and ecological space [11–13]. PLES is a comprehensive territorial spatial division, which is based on the multifunctional perspective of land use; PLES-related research is currently focused on the Chinese context, but its essence is the deepening of the research and application of land-use multifunctionality, and the related results have been widely studied at home and abroad [14–20]. Production space refers to the land used for human survival and livelihood and is an important aspect of the development of PLES; living space refers to the land used for human social habitation, which is the core of PLES; and ecological space refers to the natural environment on which human beings depend for their survival, and which is the prerequisite and direction for the development of PLES [21,22]. As a result of the evolution of the territorial system of human–land relation, exploring PLES driving mechanisms can effectively explain the relationship between human activities and the evolution of PLES patterns.

The study of PLES on the Indochina Peninsula has focused on changes in forest ecological space in each country. Studies have shown that timber export is the main economic source in Laos and Myanmar, and excessive logging has led to a decrease in the forest area and an increase in the area of grassland, arable land, and built-up land in northern Laos [23]. In Myanmar, deforestation and degradation are serious, forest area is decreasing, and poppy cultivation is increasing year by year [24]. Northern Thailand is dominated by rotational agriculture, which is gradually being commercialized as the area under monoculture cash-crop rubber forest continues to expand [9]. In addition, the conversion of forest land to cropland is very common in the Indochina Peninsula [25]. In terms of impact factor studies, previous studies have explored the driving mechanism of PLES from the aspects of physical geography, socioeconomics, the geo-environment, and trade cooperation [10], the study areas have focused on hotspots and fragile areas, and there are few studies on the driving mechanism of PLES in the Indochina Peninsula. Currently, the driving mechanism behind PLES pattern evolution is studied using multiple linear regression [26] and principal component analysis [27], but these methods only focus on the mathematical logic among factors: they cannot explain the geographic logic of the factors or the real spatial characteristics of the regression parameters; nor can they respond to the spatial heterogeneity between the dependent variables and their influencing factors in geographic phenomena [28]. Some scholars have extended ordinary linear regression by using Geodetector [29], geographically weighted regression (GWR) [30], and multiscale geographic weighted regression (MGWR) [31]; these explain the local spatial relationships

and spatial heterogeneity of variables well [32]. However, the influencing factors involved in PLES pattern evolution are both spatially and temporally nonstationary; because of this, the traditional regression and constant-coefficient spatial econometric models cannot satisfy the research need to identify the direction and strength of the drivers of PLES pattern evolution under different spatial and temporal distributions. The geographically and temporally weighted regression (GTWR) model proposed by Huang et al. [33] can effectively deal with the problem of spatiotemporal heterogeneity and has been widely used in the study of spatiotemporal heterogeneity of socio-economic and environmental pollution drivers. Based on a grid scale, this paper applies the GTWR model to the analysis of the driving mechanism of the spatial variation of territories in the Indochina Peninsula, taking into account the spatial and temporal nonstationarity of the factors and proposing a new research perspective.

This paper takes the Indochina Peninsula as the study area and explores the evolution of the spatial pattern of the Indochina Peninsula and its driving mechanism from 2010 to 2020 based on a grid scale using the land transfer matrix, the land use dynamic index, and the GTWR model. This will enrich the study of PLES on the Indochina Peninsula as a whole and inform planning for sustainable development in the area.

## 2. Materials and Methods

### 2.1. Site Description

Due to factors such as historical background, regional economic cooperation, and frequent armed conflicts, human–land relations across the Indochina Peninsula are typified by regional characteristics. The overall region is known as a fragmented zone in the world [34,35]. In this paper, Myanmar, Vietnam, Laos, Thailand, and Cambodia, all located on the Indochina Peninsula, were selected as the study area (Figure 1). As a relatively independent geographic unit, the Indochina Peninsula has an important geopolitical and economic strategic value and is an important arena for competition and power games among extra-regional powers [36,37]. The implementation of many international economic cooperation and resource development projects, especially the "Golden Four Corners" program and the "Alternative Cultivation" policy shared by China, Myanmar, Thailand, and the Lao People's Democratic Republic, brought about significant change in the land use/cover status of the region [10,38]. Together, the special characteristics of land-use patterns [23,39], tropical rain forests in Southeast Asia, and widely distributed alternative plantation crop areas [40] make the Indochina Peninsula an area of great interest to many international organizations studying LUCC and the ecological environment.

### 2.2. Data Source

Data were gathered from the GlobeLand30 land cover/land use dataset (http://www.globallandcover.com/ accessed on 2 June 2023) and the SEDAC population density dataset (https://sedac.ciesin.columbia.edu/ accessed on 4 June 2023) for 2010 and 2020. The data were preprocessed and reclassified to obtain the 2010 and 2020 PLES data for the Indochina Peninsula; the classification system is shown in Table 1 [41,42].

In this paper, road networks, water systems, population densities, night lighting, precipitation, the normalized difference vegetation index (NDVI), and armed conflict events were selected as the influencing factors for the evolution of the PLES patterns in the Indochina Peninsula region from four aspects: human location, socio-economics, natural environment, and geopolitics. Road networks affect land use in a unique way. On the one hand, road construction promotes the development of construction land; on the other hand, it takes up a large amount of forest and grassland, resulting in the reduction in forest and grassland areas. At the same time, slope greening and ecological protection undertaken in the process of road construction were shown to increase the area of shrubs and bushes [43]. In addition, proximity to a river system affects the distribution of productive space: the closer the river, the more productive the space [44]. Population density is another factor directly influencing the spatial distribution of land use/cover in

the study area, with more densely populated areas having higher levels of living-space development. Furthermore, the nighttime lighting index reflects the level of regional economic development and compensates for the lack of GDP data. Precipitation impacts the ecological space's spatial distribution, while the NDVI directly reflects the changes in forest land, grassland, cultivated land, and other land types. Finally, geopolitics is also an easily overlooked influence for the Indochina Peninsula, where the location and frequency of armed conflict events have a dramatic effect on land-type change [45]. The data sources for each influencing factor are shown in Table 2.

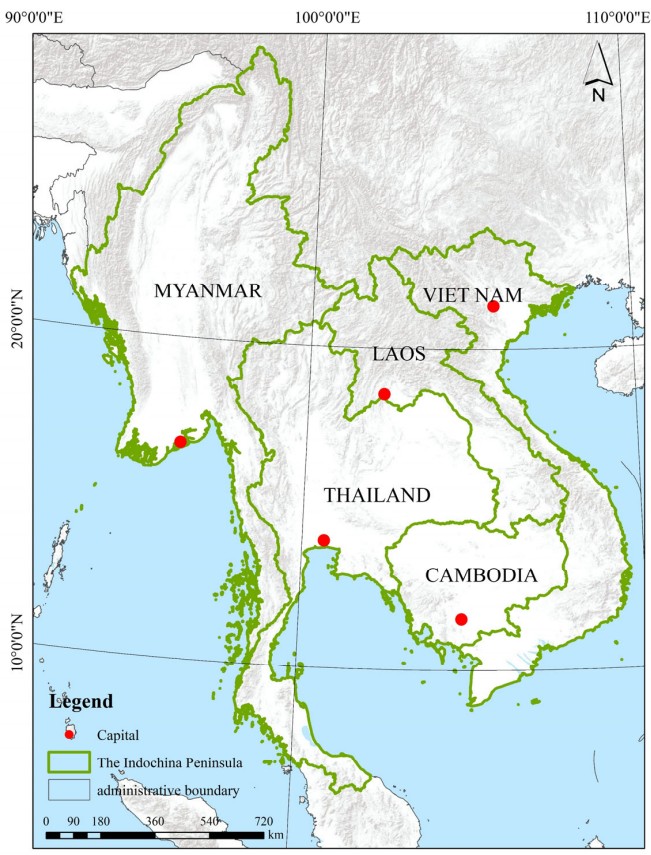

**Figure 1.** Schematic of the study area.

**Table 1.** Classification system of PLES.

| Primary Category | Secondary Category | Data Source |
| --- | --- | --- |
| The production space | 1—Agricultural production space; | GlobeLand30: cropland |
| | 2—Industrial production space; | GlobeLand30: artificial surface (excluding the range of living space defined by SEDAC) |
| The living space | 3—Urban living space; | SEDAC: the population density is greater than 1500/km$^2$ |
| | 4—Rural living space; | SEDAC: the population density is 300–1500/km$^2$ |
| The ecological space | 5—Forest ecological space; 6—Grassland ecological space; | GlobeLand30: forest, bush GlobeLand30: grass |
| | 7—Water ecological space; | GlobeLand30: wetlands, water, glaciers, and permanent snow cover |
| | 8—Other ecological spaces | GlobeLand30: tundra, bare land |

**Table 2.** Driving factors and data sources.

| Datatypes | Parameter | Factor | Introduction to Data | Data Source |
|---|---|---|---|---|
| Humanistic location | X1 | Distance to railway | Indicates the distance from the center of each pixel to the nearest railway line | https://www.openstreetmap.org (accessed on 10 July 2023) https://www.naturalearthdata.com/ (accessed on 7 July 2023) |
| | X2 | Distance to road | Indicates the distance from the center of each pixel to the nearest road | Socioeconomic Data and Applications Center | SEDAC (columbia.edu) https://www.openstreetmap.org |
| | X3 | Distance to river | Indicates the distance from the center of each pixel to the nearest river | https://www.openstreetmap.org |
| Social economy | X4 | Night lights | Indicates the nighttime light value within each pixel | VIIRS Nighttime Light (mines.edu) geodata.cn |
| | X5 | Population density | Denotes the value of the population density within each pixel | https://sedac.ciesin.columbia.edu/ (accessed on 4 June 2023) |
| Natural environment | X6 | Precipitation | Indicates the value of rainfall within each pixel | Climatic Research Unit—Groups and Centres (uea.ac.uk) |
| | X7 | Normalized difference vegetation index (NDVI) | Indicates the NDVI value within each pixel | https://ladsweb.modaps.eosdis.nasa.gov/ (accessed on 8 July 2023) |
| Geopolitics | X8 | Armed conflict events | Indicates the number of deaths from armed conflicts in each pixel. | ACLED | Bringing Clarity to Crisis (acleddata.com) |

*2.3. Data Preprocessing*

The humanistic location data require geometric repairing, cropping, merging, fusing, etc., and Euclidean distances must be calculated. Armed conflict events require data cleansing and interpolation. Compared to other interpolation methods, inverse distance weighting (IDW) is a precision interpolation and ensures that the predicted value at the sampling point is completely consistent with its real value. In this study, IDW was used to spatialize fatalities in armed conflicts. All factor data were standardized, with a uniform scale, and projected to the "Krasovsky_1940_Albers" coordinate system with a resolution of 250 m. The resolution of the PLES data was 30 m, which facilitated subsequent processing.

To verify whether initially selected influencing factors can be used for modeling the regression model, a covariance line analysis of the influence factors is required [46,47]. In this paper, the variance inflation factor (VIF), eigenvalue, etc., were selected to determine whether there was covariance among the influencing factors. When the VIF is greater than 10, it indicates that there is covariance among various factors; this must be eliminated to render the next modeling more feasible [48]. As shown in Table 3, the VIF values of the influencing factors were less than 10, so there was no multicollinearity; accordingly, they can be used for modeling the regression model.

**Table 3.** Collinearity diagnostics.

| VIF | X1 | X2 | X3 | X4 | X5 | X6 | X7 | X8 |
|---|---|---|---|---|---|---|---|---|
| 2010 | 1.125 | 1.068 | 1.069 | 1.283 | 1.326 | 1.061 | 1.104 | 1.035 |
| 2020 | 1.140 | 1.121 | 1.068 | 1.065 | 1.088 | 1.072 | 1.077 | 1.040 |
| 2010–2020 | 1.131 | 1.077 | 1.063 | 1.166 | 1.196 | 1.06 | 1.085 | 1.038 |

After preprocessing the driver data, a 15 km × 15 km fishing network was generated based on the ArcGIS platform; 8539 sampling points were obtained in the five-country region of the Indochina Peninsula. Factors were resampled uniformly up to 15 km, and factor attribute values were extracted based on the sampling points. The area and percentage of the PLES within the unit grid were counted to obtain information on the evolution of the PLES (dependent variable) and the data on the driving factors (independent variables). The driving mechanisms behind the PLES pattern evolution in the five countries of the Indochina Peninsula were analyzed using the GTWR model.

### 2.4. Research Methods

#### 2.4.1. Transfer Matrix of PLES

The land-use transition matrix was used to analyze the evolution of PLES patterns. The essence of the transfer matrix is to use the transfer probability of a Markov chain and the steady-state equation to analyze the dynamic characteristics and development trends of land-use change [49], with the following expression:

$$S_{ij} = \begin{bmatrix} S_{11} & \dots & S_{1n} \\ & \dots & \\ S_{n1} & \dots & S_{nn} \end{bmatrix} \quad (1)$$

where $S_{ij}$ is the number of transfers from spatial type $i$ to spatial type $j$ in the study area, and $S_{nn}$ is the PLES-type area.

#### 2.4.2. Land-Use Dynamics Index

Land-use dynamics refers to the quantitative change in land-use types in a certain period of time, mainly reflecting the intensity of land-use change and regional differences in the rate of change and characterizing the impact of human activities on regional land use, so as to better guide regional land use [50]. It is mainly divided into single land-use dynamics and comprehensive land-use dynamics: single land-use dynamics is used to describe the change in a certain land-use type in the region within a certain time frame, while comprehensive land-use dynamics describes the overall rate of land-use change in the entire region [51].

$$K = \frac{U_a - U_b}{U_a} \times \frac{1}{T} \times 100\% \quad (2)$$

where $K$ is the single dynamic index of a land-use type in the study period, $U_a$ is the area of a land-use type at the beginning of the study, $U_b$ is the area of the land-use type at the end of the study, and $T$ is the time interval.

$$L_c = \left[ \frac{\sum_{i=1}^{n} \Delta L_{U_{i-j}}}{2 \sum_{I=1}^{N} L_{U_i}} \right] \times \frac{1}{T} \times 100\% \quad (3)$$

where $L_c$ is the comprehensive land-use dynamics index in the study area, $L_{U_i}$ is the area of land-use type in the previous period, $\Delta L_{U_{i-j}}$ is the absolute value of the area of land of category $i$ converted to land-use type $j$ in the study time period, and $T$ is the time interval.

### 2.4.3. Standard Deviation Ellipse Model

Using the center of gravity, the long and short axes and the azimuthal angle of the standard deviation ellipse can portray the overall distribution characteristics, the degree of agglomeration, and the center of agglomeration of each type of territorial spatial area. The smaller the area of the ellipse and the standard distance between the x and y axes, the higher the degree of agglomeration, and vice versa [52]. In this study, the analysis of the standard deviation ellipse of production space, living space, and ecological space was conducted to explore the distribution range and directional trend of each space and further analyze the evolution characteristics.

$$X = \frac{1}{N} \sum_{i=1}^{n} x_i \tag{4}$$

$$Y = \frac{1}{N} \sum_{i=1}^{n} y_i \tag{5}$$

$$\tan \theta = \frac{\left(\sum_{i=1}^{n} \hat{x}_i^2 - \sum_{i=1}^{n} \hat{y}_i^2\right) + \sqrt{\left(\sum_{i=1}^{n} \hat{x}_i^2 - \sum_{i=1}^{n} \hat{y}_i^2\right)^2 + 4\left(\sum_{i=1}^{n} \hat{x}_i^2 \hat{y}_i\right)^2}}{2 \sum_{i=1}^{n} \hat{x}_l \hat{y}_i} \tag{6}$$

where $X$ and $Y$ denote the coordinates of the center of gravity position of the spatial unit of the land-use type, $x_i$ and $y_i$ denote the value of the coordinates of the spatial unit, $\theta$ denotes the angle of the ellipse, $\hat{x}_i$ and $\hat{y}_i$ denote the deviation from the center coordinates to the center of gravity coordinates of each spatial unit, respectively. The main parameters of a standard deviation ellipse are the position of the center point, the long axis, the short axis, and the angle of rotation.

### 2.4.4. GTWR Model

GTWR is a regression analysis method that incorporates temporal and spatial information on the basis of an ordinary linear regression (OLR) to study spatiotemporal heterogeneity; it is capable of reflecting the change patterns in the spatiotemporal non-stationarity of the regression coefficients [33]. The formula is:

$$y_i = \beta_0(u_i, v_i, t_i) + \sum_k \beta_k(u_i, v_i, t_i) X_{ik} + \varepsilon_i \tag{7}$$

where $(u_i, v_i)$ denotes the latitude and longitude coordinates of the $i$th sample point, $t_i$ denotes the time of observation, $y_i$ denotes the value of the dependent variable for the ith sample point, and $X_{ik}$ denotes the $k$th explanatory variable for the $i$th sample point. $\varepsilon_i$ is the model error term, $\beta_0(u_i, v_i, t_i)$ denotes the regression constant for the $i$th sample point, and $\beta_k(u_i, v_i, t_i)$ denotes the regression coefficient of the $k$th explanatory variable for the $i$th sample point. This is expressed as follows:

$$\hat{\beta}(u_i, v_i, t_i) = \left[X^T W(u_i, v_i, t_i) X\right]^{-1} X^T W(u_i, v_i, t_i) Y \tag{8}$$

where $W(u_i, v_i, t_i)$ denotes the weight of spatiotemporal location $i$. The GTWR model determines the weight of the influence of the values of other sample points on the regression sample points by constructing a spatiotemporal weight matrix.

The process of constructing a weight matrix based on spatiotemporal distance is shown below. First, the spatial distance between the sample points is calculated by applying the Euclidean distance formula.

Since the different units of measurement for temporal and spatial distances are prone to affect the results, the temporal and spatial distances are calculated as follows:

$$(d^{ST})^2 = \lambda(d^S)^2 + \mu(d^T)^2 \tag{9}$$

The weight function is usually chosen as either a Gaussian or a bisquare function; these can be transformed into a weight function after substitution, and the weight matrix is calculated as follows:

$$W_{\mathrm{ij}} = exp\{-\frac{[(u_i - u_j)^2 + (v_i - v_j)^2] + \tau(t_i - t_j)^2}{(h_s)^2}\}$$

(10)

where $\tau = \mu/\lambda, \mu, \lambda$ are the weights used to balance the different effects, $d_{ij}^{ST}$ is the spatiotemporal distance between the sample points, and $h$ is a non-negative parameter called the spatiotemporal bandwidth.

## 3. Results

### 3.1. Analysis of the Dynamics of Spatiotemporal Patterns in PLES

Ecological space dominates the PLES of the Indochina Peninsula. In the 10-year period from 2010 to 2020, the areas of production and living spaces increased dramatically, while the area of ecological space decreased correspondingly; the trend in change is consistent with the characteristics of the regional resources and economic development (Figure 2). From the point of view of changes in the area of each type of space, with population growth and economic development, both urban and rural living spaces expanded, with additional areas of 3460 and 2029 km$^2$ in the 10-year period, respectively. The Indochina Peninsula is relatively backward in terms of economy and industry but has developed its agriculture. Traditional means of farming such as slash-and-burn and straw burning, combined with the expansion of plantations and economic forests, such as rubber forests, commercial logging, land reclamation and regional economic cooperation, have contributed to the rapid expansion of agricultural and industrial production spaces, while the ecological spaces of woodland and grassland have decreased at different levels, with woodland decreasing by 26,549 km$^2$ and grassland decreasing by 3624 km$^2$, making woodland ecological space the land-use type with the greatest change in area on the Indochina Peninsula.

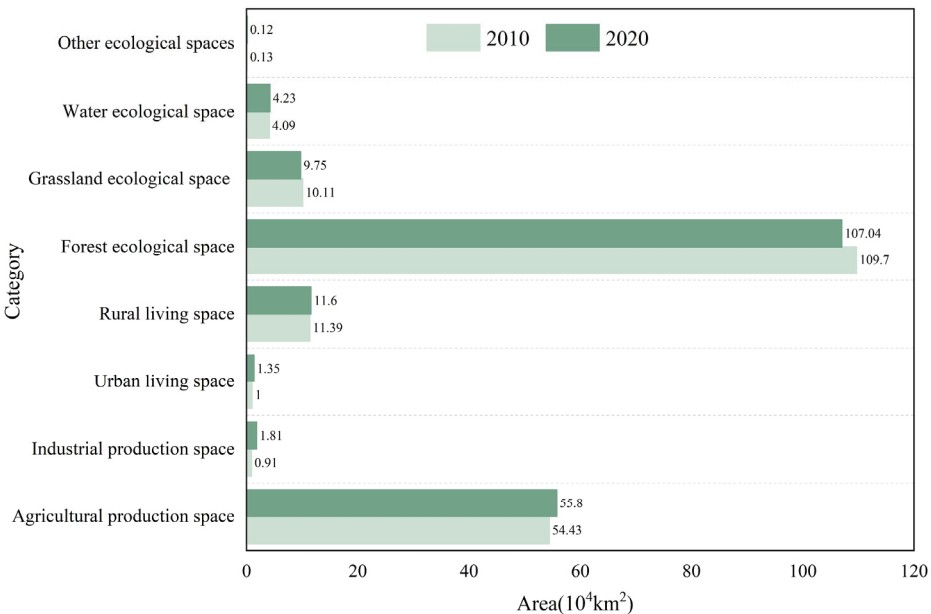

**Figure 2.** Land-use dynamics index for PLES on the Indochina Peninsula.

There are spatial differences in the rate of change in PLES in the Indochina Peninsula region (Table 4). From 2010 to 2020, the integrated land-use dynamics of the Indochina Peninsula was 0.16%, and those of Myanmar, Vietnam, Laos, Cambodia, and Thailand were 0.07%, 0.3%, 0.14%, 0.71%, and 0.13%, respectively. Cambodia had the fastest rate of change in the spatial pattern of PLES, Laos the next fastest, and Myanmar the slowest. The rate of

spatial pattern change was influenced by regional economic development. The Indochina Peninsula experienced a rapid expansion in industrial production space with a motivation of 9.84%, followed by urban living space with a motivation of 3.44%. Rural living space was relatively stable in area, having a low growth rate with a motivation of 0.18%. Forest land, grassland, and other ecological spaces had a motivation of −0.24%, −0.36%, and −0.42%, respectively, with other ecological spaces decreasing at the fastest rate.

**Table 4.** Land-use dynamics index for PLES in the five countries on the Indochina Peninsula.

| ID | Category | Laos | Cambodia | Myanmar | Thailand | Vietnam | The Indochina Peninsula |
|----|----------|------|----------|---------|----------|---------|-------------------------|
| 1 | Agricultural production space | 0.80% | 1.97% | −0.14% | 0.01% | 0.47% | 0.25% |
| 2 | Industrial production space | 63.19% | 7.56% | 5.50% | 13.69% | 10.85% | 9.84% |
| 3 | Urban living space | 2.53% | 6.63% | −0.42% | 7.45% | 2.12% | 3.44% |
| 4 | Rural living space | 3.14% | 1.12% | 1.64% | −1.59% | 0.42% | 0.18% |
| 5 | Forest ecological space | −0.18% | −1.23% | −0.05% | 0.00% | −0.55% | −0.24% |
| 6 | Grassland ecological space | 0.10% | −0.70% | −0.02% | −1.05% | −0.58% | −0.36% |
| 7 | Water ecological space | 1.96% | 0.42% | 0.37% | −0.51% | 1.57% | 0.34% |
| 8 | Other ecological spaces | 1.40% |  | −2.62% | 422.08% | 8.21% | −0.42% |
|  | Comprehensive land-use dynamic index | 0.14% | 0.71% | 0.07% | 0.13% | 0.30% | 0.16% |

From 2010 to 2020, the industrial production space of the Lao PDR was expected to change at the highest rate of 63.19%, following the Lao Government's active promotion of the strategy for "resources for capital". Thailand's other ecological spaces were expected to undergo drastic changes, with a 422.08% change in dynamics; this was mainly influenced by Thailand's national development strategy driven by commercial logging, urban development, and the acquisition of international benefits [53].

*3.2. Spatiotemporal Analysis of the Evolutionary Process of PLES*

3.2.1. Quantitative Analysis of Land-Use-Type Shifts in PLES

From 2010 to 2020, the Indochina Peninsula had an area of 212,818.70 km$^2$ of interconversion of PLES utilization, manifested in the conversion of ecological space into production space, and the interconversion of woodland and grassland ecological spaces (Figure 3). The transfer in of industrial production space came predominantly from agricultural production space, accounting for 49.48% of the industrial production land by 2020. Transfer out was mainly converted to agricultural production space, accounting for 76.11% of the total transfer out of industrial production. The change in woodland ecological space was predominantly attributed to grassland ecological and agricultural production spaces; by 2020, the transfer in accounted for 3.62% and 1.87% of the area of woodland ecological space, respectively. The area of forest ecological space converted to grassland ecological and agricultural production spaces was 39,470 km$^2$ and 41,370 km$^2$, respectively, accounting for 43.69% and 45.79% of the total transfer out of forest land. The water ecological space was predominantly converted to agricultural production space, accounting for 54.21% of the transfer out of water ecological space.

Agricultural production space and woodland ecological space are the main land types in the five countries of the Indochina Peninsula, and from 2010 to 2020, there were shifts in the various types of PLES. In Myanmar and Laos, the most drastic land transfer of three biospatial land types was the interconversion of woodland and grassland ecological spaces, followed by the interconversion of woodland ecological and agricultural production spaces. Thailand's land transfer of PLES mainly focused on the interconversion of woodland ecological and agricultural production spaces, being 8334.99 km$^2$ and 7700.08 km$^2$, respectively. The interconversion of three biospatial land types in Cambodia and Vietnam focused on the transfer of woodland ecological to agricultural production spaces. The quantitative transfer mainly focused on the conversion of woodland ecological space into agricultural production and grassland ecological spaces, with the areas of woodland

converted to agriculture being 11,810.08 km² and 9744.81 km², respectively. The areas of land converted to grassland were 2551 km² and 7455.21 km², respectively. The reason for this may be that changes in various types of land areas from expansion to contraction, or from contraction to expansion, corresponded to the transformation in the stage of regional economic development [54]. With the construction of the regional economic corridor, the five countries of the Indochina Peninsula changed from traditional agricultural methods to commercial agricultural production, and the center of gravity tilted from the primary industry to the secondary and tertiary industries: this had a direct impact on the transformation of ecological and production spaces.

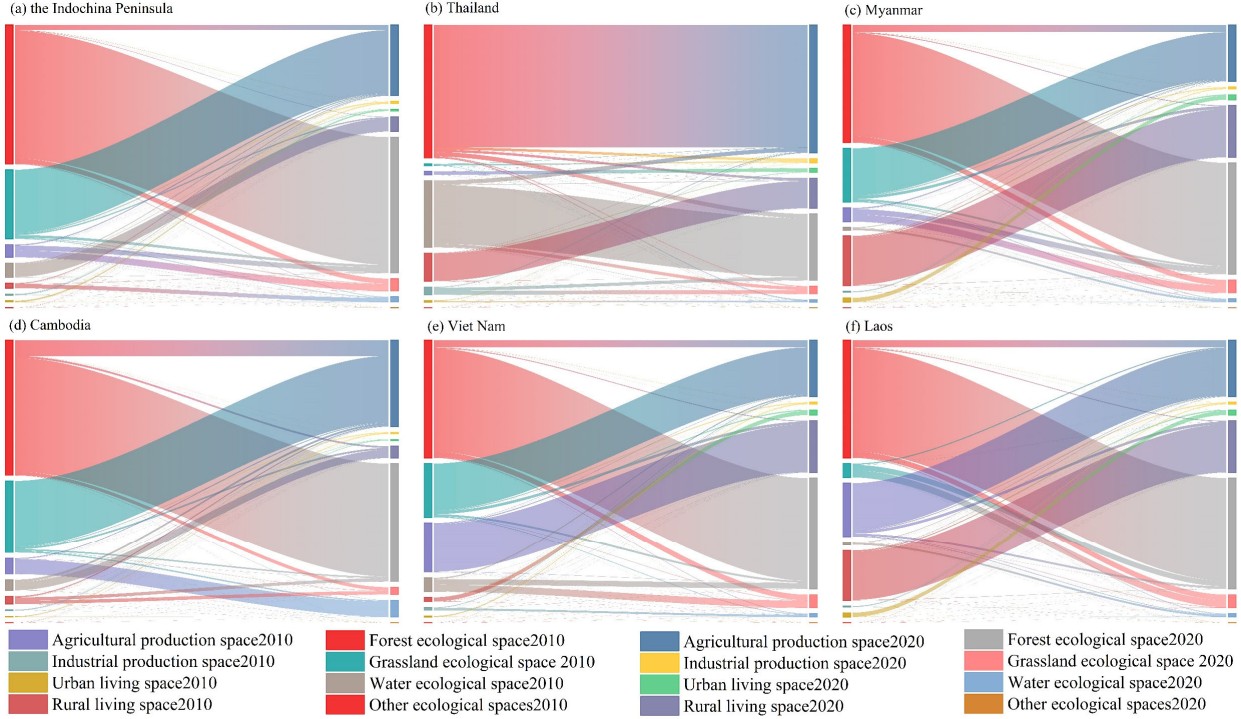

**Figure 3.** Amount of land-use-type transfers in PLES from 2010 to 2020. (**a**) Indochina Peninsula, (**b**) Thailand, (**c**) Myanmar, (**d**) Cambodia, (**e**) Vietnam, (**f**) Laos.

### 3.2.2. Analysis of the Process of Transferring Land-Use Types in PLES

The conversion of production and ecological spaces is distributed as a network throughout the Indochina Peninsula, and the conversion of living space is distributed as points (Figure 4). The conversion of forest and grassland ecological spaces is distributed in the western region of Myanmar, the northeastern and southern regions of Thailand, and the entire territories of Vietnam and Cambodia. From 2010 to 2020, the conversion of forest ecological space into agricultural production space was uniformly distributed across the Indochina Peninsula. The conversion of agricultural production space into rural living space was distributed in a pointlike manner in the southern region of Yangon and the central region of the Sagaing region in central Myanmar, Chonburi Province in Thailand, Batuyi, Battambang, and Kandan provinces in Cambodia, Binh Duong, Đồng Nai, Bạc Liêu, and Sóc Trăng provinces in Vietnam, and the western region of Laos. The conversion of agricultural production space into industrial production space was distributed in Myanmar, Vietnam, central Thailand, and central Cambodia, but with almost none in Laos. The conversion of grassland ecological space into agricultural production space was concentrated throughout the eastern and central regions of the Indochina Peninsula, with very little distribution in the southwestern region. The conversion of rural living space into agricultural production space was concentrated in the central region of Thailand and the southeastern region of Cambodia. The conversion of the water ecological space and agricultural production space into each other was evenly distributed across the Indochina Peninsula. The exception

to this was Laos, where the main food crop is rice: some cultivated land and forested land were converted into agricultural production space in order to improve agricultural production conditions. Rice is the main food crop across the Indochina Peninsula, and in order to improve agricultural production conditions, some areas of cultivated and forest land have been converted into paddies. The conversion of industrial production space into agricultural production space is discretely distributed across the Indochina Peninsula from northwest to southeast in the middle of Myanmar, Cambodia, and Thailand.

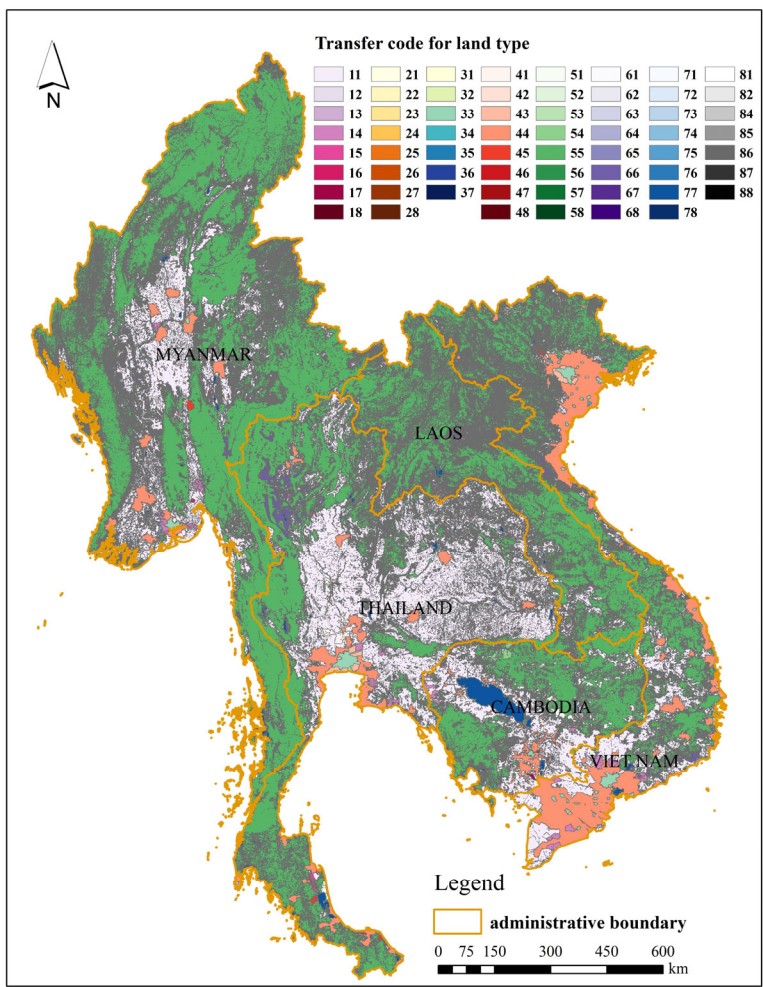

**Figure 4.** Patterns of PLES on the Indochina Peninsula from 2010 to 2020.

The long half-axis of the ellipse indicates the direction of the data distribution, and the short half-axis indicates the range of the data distribution; the larger the difference between the values of the long and short half-axes, the more obvious the direction of the data is [52]. The migration path of the center of gravity of PLES on the Indochina Peninsula demonstrates significant directional differences, and the long semi-axis is geographically oriented north–south with both Laos and Vietnam (Figure 5). Overall, from 2010 to 2020, the production space migrated to the southwest, the living space to the northeast, and the ecological space to the east (Table 5). Compared with the living space, the production and ecological spaces experienced a more directional tendency, were more influenced by Myanmar, Thailand, and Cambodia, and exhibited a more centralized trend in the distribution of ecospatial data. PLES migrated to the southwest, and the center of gravity of the production space was located in the north of Thailand and moved insignificantly, the standard deviation ellipse area did not change much, the X-axis decreased, and the Y-axis increased, indicating that the production space distribution across the Indochina Peninsula was more balanced, and mainly in the northwest–southeast direction. The center of gravity

of the living space was located in the east of Thailand, shifting towards the northeast, and the standard deviation ellipse area for the living space decreased in both the *X*- and *Y*-axes, indicating that the living space had a tendency to shrink in all directions; this shows a tendency to shift from a discrete to an agglomerated distribution, this phenomenon being mainly concentrated in the regions of Thailand, Laos, and Vietnam. The direction and extent of the standard deviation ellipse distribution of the ecological space was similar to that of the production space, indicating that the trend in the living space was stronger in the northwest–southeast direction than in the northeast–southwest direction. The center of gravity of the ecological space was located in the middle of Thailand and shifted to the east, and the direction and extent of the standard deviation ellipse distribution was similar to that of the production space, indicating that the development trend of the living space was stronger in the northwestern–southeastern direction than in the northeastern–southwestern direction.

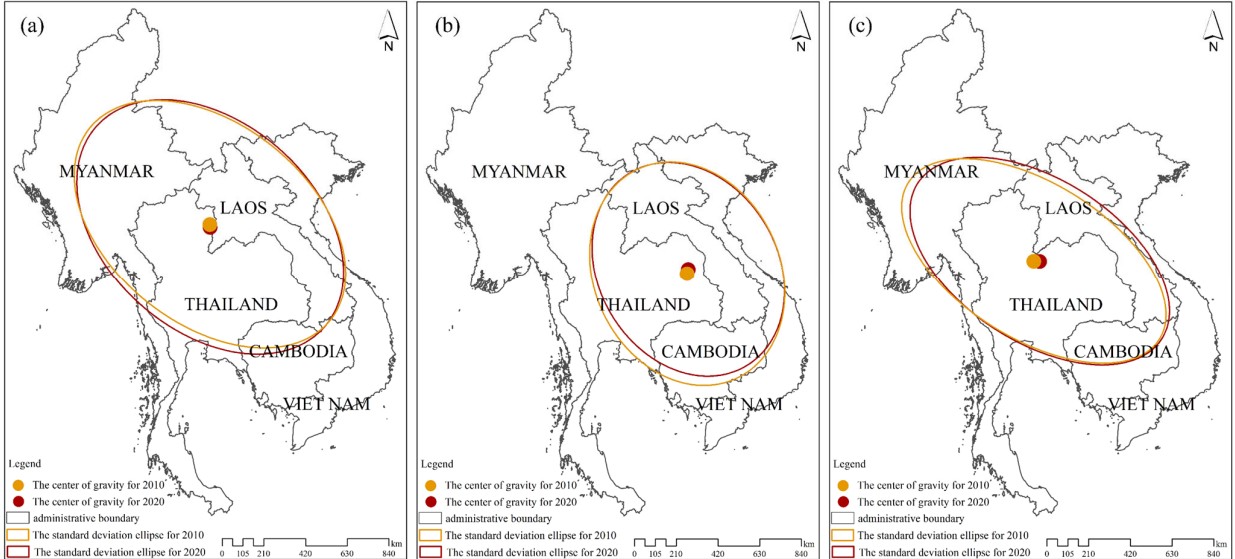

**Figure 5.** The standard deviation ellipse of PLES on the Indochina Peninsula from 2010 to 2020. (**a**) The production space, (**b**) the living space, and (**c**) the ecological space.

**Table 5.** Elliptic standard deviation parameter.

| Id | Category | Center *X* | Center *Y* |
|----|----------|-----------|-----------|
| 1 | The production space 2020 | 101.075225 | 18.664372 |
| 2 | The production space 2010 | 101.064734 | 18.789373 |
| 3 | The living space 2020 | 104.048867 | 16.76586 |
| 4 | The living space 2010 | 104.001003 | 16.56669 |
| 5 | The ecological space 2020 | 101.289046 | 17.13435 |
| 6 | The ecological space 2010 | 101.024312 | 17.142921 |

### 3.3. Analysis of PLES Spatiotemporal Pattern Evolution Drivers

In this study, road networks, water systems, population densities, night lighting, precipitation, NDVI, and armed conflict events were selected as the influencing factors in the evolution of PLES spatiotemporal patterns in the Indochina Peninsula region from the four aspects of humanistic location, socio-economics, natural environment, and geopolitics. Covariance diagnostics and standardization were performed on the influencing factors, and the results demonstrated that all the factors satisfied the model construction criteria. The GTWR model was applied to a regression analysis of the sample data to obtain the regression coefficients of each influencing factor on the evolution of the PLES pattern, based on the grid scale from 2010 to 2020. The model was used to analyze the degree

of influence of each factor on the evolution of the PLES pattern at different spatial and temporal locations under the double effect of time and space. The GTWR model was applied to simulate the eight spatial types: different R2 and bandwidths were obtained, with an optimal model fit of 0.56 and a mean value of 0.4, the lowest fit being that of the watershed ecological space. The magnitude of the regression coefficients represents the degree of influence of each influencing factor on the evolution of the three spatial patterns (Table 6). The transfer of land-use types in the PLES of the Indochina Peninsula was influenced by social context and regional environment. Population density (X5) was the factor that most influenced the changes in pattern of the three living spaces; regions with a high population density were prone to an expansion of production and production space, and ecological space was prone to being squeezed. In 2020, the factors influencing the agricultural production space and the ecological space of the forest land had opposing roles. Increased population density promoted the development of agricultural production space, while inhibiting the development of woodland ecological space. The armed conflict factor (X8) had a positive feedback effect on urban living space and inhibited the conversion of other spatial types into the promotion of agricultural production space, woodland, and other ecological space. This is because the political and ethnic conflicts in Myanmar, as well as turbulence in Thailand's political environment, and the potential for social instability, etc., intensified the outbreak of armed conflict events to a certain extent, affecting the environment of human life and production. Furthermore, in 2020, possibly because of the move to promote agricultural production space, it appears that agriculture was not affected by the waves of armed conflict. Instead, agriculture production space played a facilitating role to a certain extent. The distance to the road network (X1, X2) factor was positively related to the ecological space of the woodland, which may be due to the increase in green environments such as street trees, shrubs, and grasses on both verges on the sides of the road. Border road construction improves accessibility, but road planning and construction also encroach on productive living space to some extent. The distance from the water system factor (X3) and precipitation (X6) were positively proportional to the ecological space of forest land, and inversely proportional to other spatial types. The Indochina Peninsula is rich in precipitation, has a dense water network, is rich in forest resources, and tropical rainforest occupies a wide range of areas; however, this inhibits the expansion of production space. Night lighting (X4) was proportional to the relationship between industrial production space and human life space, reflecting the regional economic level: the higher the level of economic development, the more frequent the human activities, and the closer to the urban built-up area. NDVI (X7) reflects the vegetation cover, which was positively proportional to the ecological space of the forest land and grassland; an increase in the vegetation cover indicates the expansion of the ecological space of forest land and grassland.

The evolution of the spatial pattern of agricultural production was affected by factors with significant spatial and temporal heterogeneities (Figure 6). The influence of each factor on the spatial quantitative changes in agricultural production created both positive and negative spatial distributions. Factor X3 was mainly positively related to the spatial relationship of agricultural production on the Indochina Peninsula, but negatively related to the spatial relationship of agricultural production in southern Myanmar, northwestern Thailand, and northern Laos; the positive feedback expanded northward over time. In 2020, factor X2 showed a large area of negative feedback; in 2010, however, there had been positive feedback in the cities of northern and southern Vietnam. By 2020, positive feedback was only evident in the cities of northern Vietnam. In 2010, factor X1 had an inverse effect on the spatial quantity change in agricultural production; in that year, it was distributed in the south of Laos and the center of Vietnam; by 2020, it had spread southward to the central and southern cities of Laos and Vietnam. For factor X5, there was little change in the distribution areas of the positive and negative effects; the negative feedback areas were distributed in the south of Myanmar, Cambodia, and Thailand, as well as in Vietnam. Factor X6 demonstrated a negative feedback area distributed in the

center of Thailand, Myanmar, Vietnam, Laos, and the northern region of Cambodia. The positive feedback area grew from 2010 to 2020 and was focused on the central region of the Indochina Peninsula. Factor X8 changes showed a decreasing trend in the negative feedback areas, with decreasing areas concentrated in Vietnam and southern Thailand; the negative feedback area in central and northern Myanmar remained almost unchanged. The negative feedback area for the nighttime lighting factor decreased, with the area in northern Myanmar and Laos decreasing, and the negative feedback area in Cambodia moving to the south. The negative feedback area for factor X7 was larger in size, and the positive feedback area was concentrated in northern Myanmar, the distribution area of negative feedback gradually decreasing in the period from 2010 to 2020.

**Table 6.** GTWR model estimation result.

| Year | Category | X1 | X2 | X3 | X4 | X5 | X6 | X7 | X8 |
|---|---|---|---|---|---|---|---|---|---|
| 2010 | Agricultural production space | −0.264 | −0.169 | 0.011 | −3.810 | 4.692 | −0.250 | −0.478 | −0.286 |
|  | Industrial production space | −0.021 | −0.034 | −0.010 | 0.038 | 0.617 | −0.002 | −0.018 | −0.063 |
|  | Urban living space | −0.127 | −0.119 | −0.092 | −1.857 | 6.327 | −0.079 | −0.038 | 1.049 |
|  | Rural living space | −0.001 | −0.043 | −0.012 | 0.626 | 1.574 | −0.023 | −0.007 | −0.075 |
|  | Forest ecological space | 0.384 | 0.314 | 0.158 | 3.360 | −13.91 | 0.344 | 0.551 | −0.203 |
|  | Grassland ecological space | 0.045 | −0.012 | −0.033 | −0.710 | 0.056 | −0.033 | 0.019 | −0.280 |
|  | Water ecological space | −0.002 | 0.060 | −0.028 | 0.539 | −0.142 | 0.000 | −0.044 | −0.121 |
|  | Other ecological spaces | 0.000 | 0.002 | −0.003 | 0.016 | 0.043 | −0.002 | −0.002 | −0.005 |
| 2020 | Agricultural production space | −0.301 | −0.551 | −0.013 | −6.364 | 5.143 | −0.191 | −0.448 | 1.449 |
|  | Industrial production space | −0.024 | −0.081 | −0.008 | 0.346 | 0.623 | −0.003 | −0.021 | −0.044 |
|  | Urban living space | −0.106 | −0.291 | −0.061 | 2.112 | 6.758 | 0.041 | −0.038 | −2.406 |
|  | Rural living space | −0.001 | −0.059 | −0.007 | 1.461 | 1.575 | −0.021 | −0.009 | −0.151 |
|  | Forest ecological space | 0.409 | 0.726 | 0.126 | 4.707 | −16.17 | 0.161 | 0.561 | 0.653 |
|  | Grassland ecological space | 0.056 | −0.002 | −0.023 | −1.960 | 0.604 | −0.006 | 0.012 | −0.095 |
|  | Water ecological space | −0.009 | 0.112 | −0.021 | −0.546 | 0.269 | −0.003 | −0.043 | −0.152 |
|  | Other ecological spaces | 0.000 | 0.001 | −0.001 | 0.010 | 0.045 | −0.001 | −0.002 | 0.001 |

Factor X1 was mainly positively related to the forest ecological space; the negative feedback areas for this factor decreased with time, the decreased areas being concentrated in the northern region of Thailand and on the border of Myanmar (Figure 7). Factor X3 was mainly positively related to the ecological space of the forest land; the negative feedback areas for factor X3 were concentrated in the central region of Myanmar and the eastern region of Thailand and decreased to the northeast with the change over time. Factor X2 experienced a decrease in negative feedback areas with the change over time. In 2020, the negative feedback areas for factor X2 were mainly in Cambodia's Battambang Province and Siem Reap Province, and Thailand's Surat Thani. The factor X5 positive feedback areas shifted from the provinces of Kandal and Takeo in Cambodia to Bangkok in Thailand. The negative feedback areas related to factor X6 spread out in all directions, and the positive feedback areas were concentrated on the Indochina Peninsula, rather than the center. Positive feedback areas for factor X8 were to the west of the Indochina Peninsula, while negative feedback areas were to the east. The negative feedback areas for this factor spread from the southeast to the northwest, being concentrated in the southern region of Myanmar. Factor X4 positive feedback areas were larger, and negative feedback areas expanded northwards. Positive feedback areas for factor X7 expanded northwards, and negative feedback areas were concentrated in the southern region of Laos. Factor X8 positive feedback areas spread northwards, while negative feedback areas were concentrated in the southern region of Laos. Positive feedback areas for factor X9 spread northwards. The negative feedback regions were concentrated in Phôngsali, Laos and the southern region of Burma.

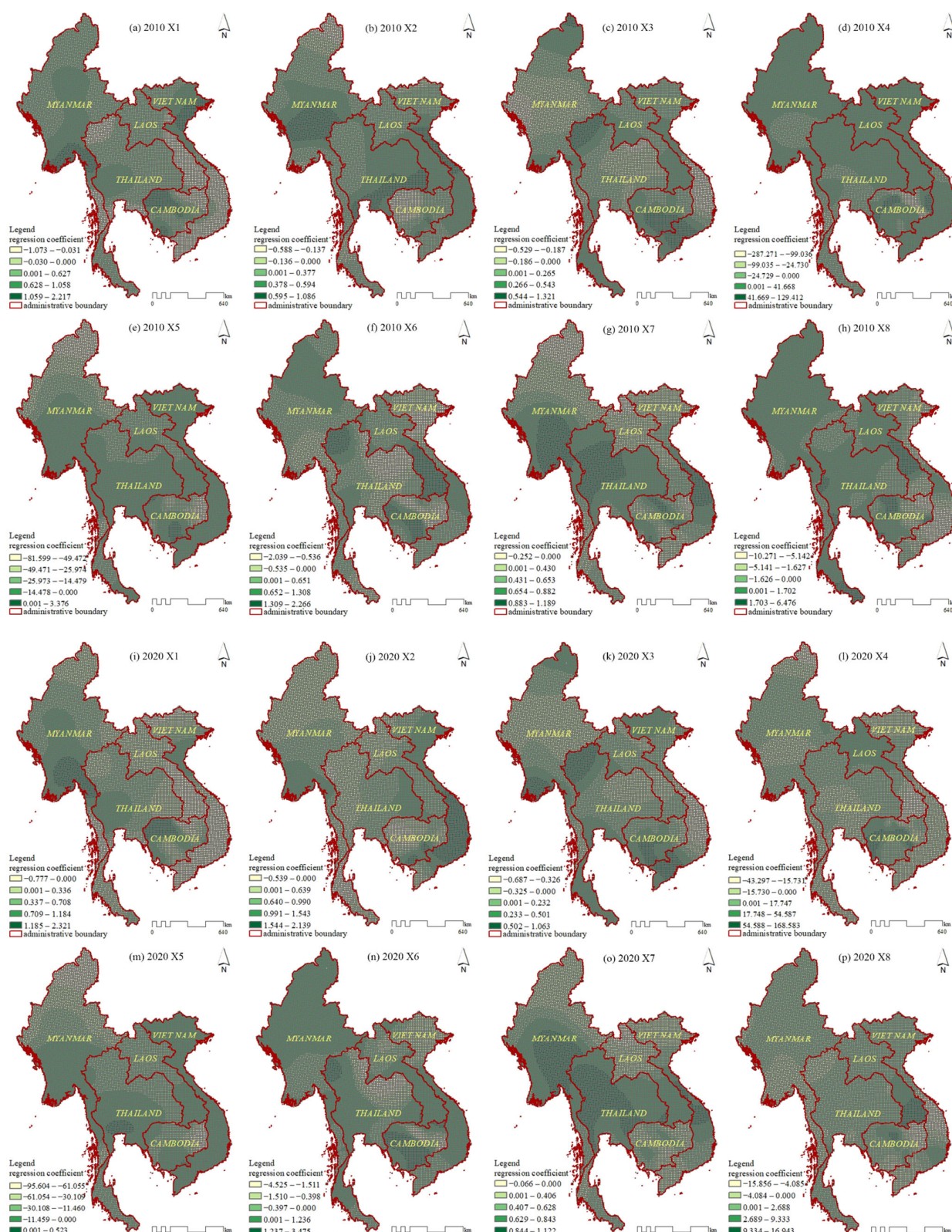

**Figure 6.** Spatiotemporal distribution of the effects of factors on agricultural production space. (**a**–**p**) represents the influence degree of different factors on the evolution of different land-use space in different periods.

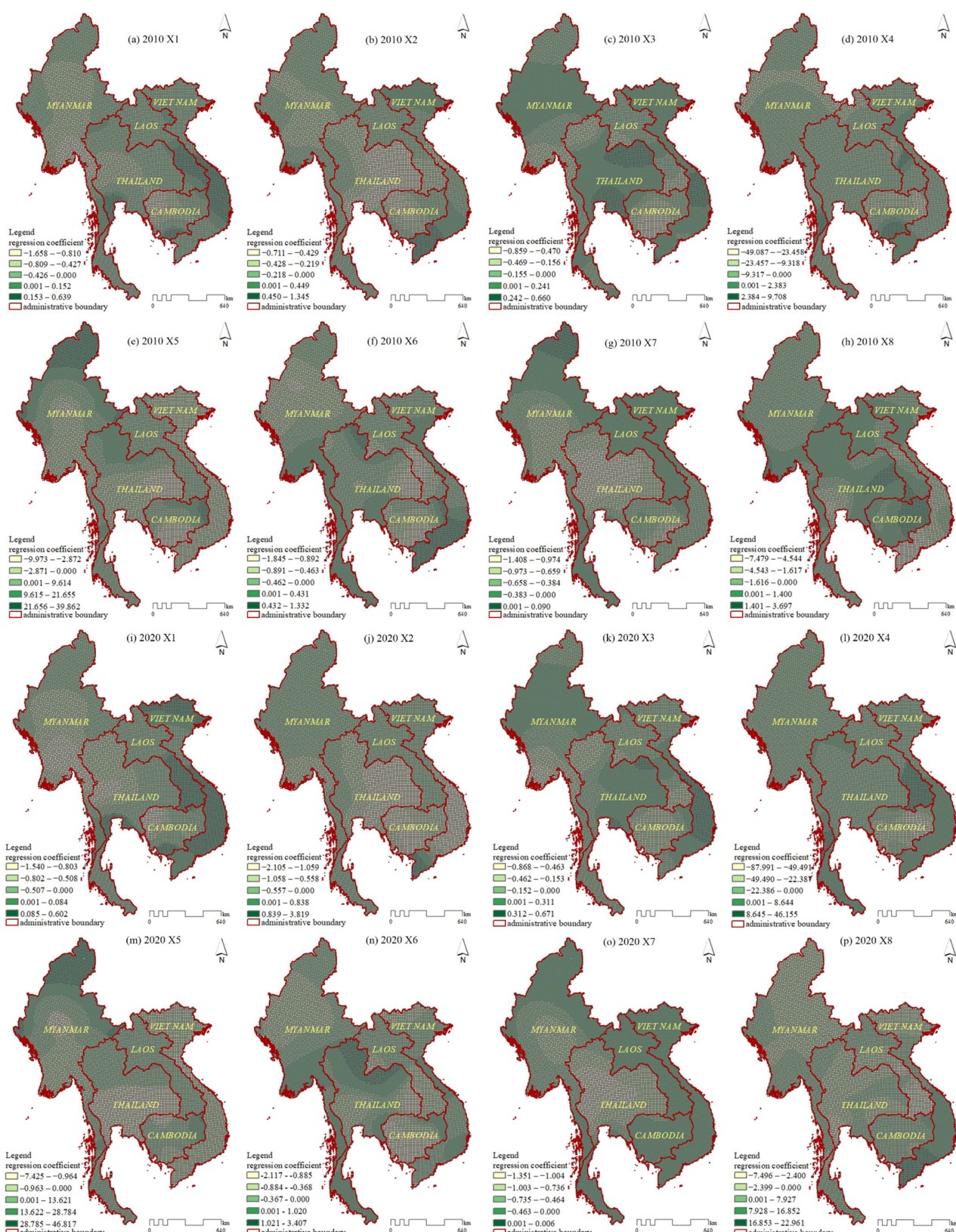

**Figure 7.** Spatiotemporal distribution of the effects of factors on forest ecological space. (**a–p**) represents the influence degree of different factors on the evolution of different land-use space in different periods.

Factor X1 led to a large change in the positive feedback area for industrial production space; this mainly occurred in Thailand, with a decrease in the distribution in the northwest and a concentration in the east (Figure A1). The change resulting from X2 was a decrease in the positive feedback areas in the central region of Vietnam and the western region of Thailand. The changes in the distribution of the X3 factor were a shift in the positive feedback area from the periphery to the middle of the Indochina Peninsula and a decrease in the northern region of the positive feedback area in Vietnam. The change resulting from factor X4 was the 2020 conversion of negative feedback to positive feedback in northern Myanmar, northern Thailand, and southern Yunnan. The changes in factor X5 were the shrinkage of positive feedback in Myanmar and Thailand to the northeast, and the expansion of positive feedback in Cambodia to the west. The change resulting from factor X6 was the expansion of positive feedback to the south. The changes created by factor X7 were the shrinkage of positive feedback in northern Myanmar, the expansion of positive feedback to the north in Thailand, and the addition of positive feedback in southern Vietnam. The change created by factor X8 demonstrates that the positive feedback areas spread from the center to the east and west.

In Myanmar and Cambodia, the influence of X1 on the negative feedback of grassland ecological space changed greatly. The distribution in Myanmar changed from the central region to the western and eastern regions, while the negative feedback area in the eastern region of Cambodia decreased (Figure A2). The change brought by the X2 factor was that the negative feedback region spread from the central region to the surrounding region. By 2020, the spatial change created by X3 was that a new positive feedback area was added in the southern region of Laos, while the change wrought by X4 was that the negative feedback region expanded to the eastern region of the country. In 2020, positive feedback areas on grassland ecological space for factor X5 were mainly concentrated in northeast Myanmar and Laos, and positive feedback areas appeared in northern Cambodia. The change resulting from factor X6 was that positive feedback areas developed from being discrete to being clustered in Cambodia and Laos, while the change created by X7 was that positive feedback areas spread from the perimeter to the center. By 2020, the change brought by X8 was positive feedback areas appearing in southern Myanmar and northern Vietnam.

In terms of rural living space, positive feedback areas for factor X1 all increased on the Indochina Peninsula (Figure A3). The change resulting from factor X2 was that positive feedback areas in Thailand expanded to the northeast, while the change created by factor X3 was that positive feedback areas expanded to the northwest. For factor X4, the change was that positive feedback areas expanded to the northeast. The factor X5 change was that negative feedback areas in Thailand expanded in a fan shape to the northeast; for factor X6, negative feedback areas in northern Vietnam converted to positive feedback areas. For factor X7, the change was that positive feedback areas in northern Thailand converted to negative feedback areas. For factor X8, the change was that negative feedback areas in northern Vietnam changed to positive feedback areas in northern Thailand. The change was that the negative feedback region in northern Vietnam converted to a positive feedback region, while in northern Thailand, it was the positive feedback region that changed to a negative feedback region. The X7 change was that the positive feedback region in Myanmar expanded to the south, while the positive feedback region in Thailand decreased to the south. The X8 change was larger, with the positive feedback regions in Myanmar and Laos shifting to the northeast, and the positive feedback region in Vietnam disappearing.

In 2010, positive feedback for X1 on urban living space was distributed in Thailand, northern Vietnam, and central Laos, and by 2020, also in Cambodia (Figure A4). Factor X2 still resulted in negative feedback, although negative feedback in Myanmar and Cambodia had weakened. The change resulting from X3 was that the negative feedback areas in northern Thailand and central Laos converted to positive feedback, and the change created by X4 was that the negative feedback areas in northern Laos and Cambodia also converted to positive feedback. Factor X5 did not create significant change, with all areas still showing positive feedback. Factor X6 caused a change as the positive feedback areas in Thailand

decreased to the northeast. The positive feedback areas in the southern cities of Vietnam expanded to the northeast. The change caused by factor X7 was that the positive feedback area in Vietnam spread from the center to the north and south, while that caused by factor X8 was that the positive feedback area spread to the center.

In terms of the watershed ecological space, the X1 positive feedback influence expanded from the northeast to the southwest (Figure A5). The changes related to the X2 factor were that the negative feedback area in Myanmar spread to the east, while the positive feedback area in Thailand expanded to the west. For X3, the positive feedback area expanded to the east. The change associated with factor X4 was that the positive feedback area in Myanmar transformed from being dispersed in the surroundings to being clustered in the center, while the change in Cambodia and Myanmar was the opposite. For factor X5, the change was that the positive feedback area spread to the southwest. One X6 factor change was that the negative feedback areas in Myanmar and Laos spread to the north. The other X6 factor change for these countries was that the positive feedback regions in Myanmar and Laos also expanded to the north. For factor X7, the change was that the positive feedback area narrowed downward to the north. The X8 factor negative feedback region spread to the southwest of Myanmar and Thailand.

Regarding the other ecological space, the X1 impact was on the expansion of the positive feedback area in Thailand in 2020 (Figure A6). The changes associated with X3 were the spread of the positive feedback area in Myanmar to the north, and the expansion of the positive feedback area in Vietnam to the south. The changes brought by X4 were the conversion of positive feedback to negative feedback in the south of Vietnam, and the change from the negative feedback area to a positive feedback area in the north; the positive feedback area in Myanmar spread to the south. For X5, the positive feedback area in Cambodia disappeared, and the positive feedback areas in Myanmar were concentrated in the center. The X6 positive feedback area spread to the southwest, with the X6 positive feedback region spreading to the south. The X7 positive feedback region spread to the southwest. The X8 positive feedback region expanded from the north of Myanmar and the south of Laos to the south, while the negative feedback region in Thailand converted to a positive feedback region.

## 4. Discussion

PLES is a comprehensive territorial spatial division, which is based on the multifunctional perspective of land use; PLES-related research is currently focused on the Chinese context, but its essence is the deepening of the research and application of land-use multifunctionality, and the related results have been widely studied at home and abroad [14–20]. Previous studies have mainly focused on LUCC simulation and prediction using different remote sensing data products and on improving the simulation accuracy to accurately assess the historical dynamics and future changes of LUCC in the region [5–8]. Studies of PLES in the region have focused on ecological spatial and borderland land-use changes in national woodlands, at the national scale, with few analyses of the Indochina Peninsula as a whole. Studies have shown that timber export is the main economic source in Laos and Myanmar, and excessive logging has led to a decrease in the forest area and an increase in the area of grassland, arable land, and built-up land in northern Laos [23]. Northern Thailand is dominated by rotational agriculture, which is gradually being commercialized as the area under monoculture cash-crop rubber forest continues to expand [9]. In addition, the conversion of forest land to cropland is very common in the Indochina Peninsula [25]. This study also found that the Indochina Peninsula as a whole is experiencing a decrease in the area of forest ecological space and an increase in the area of living space and productive space, indicating that the area of forested land is decreasing while the area of land types such as cropland and built-up land is increasing. These changes are mainly caused by economic policies and demographic, socio-economic, and environmental changes [6,55], which are similar to the drivers of this paper. The difference is that the natural environmental factors in this paper only considered precipitation and NDVI and did not consider other

environmental factors such as soil properties and light radiation. In addition, this paper incorporated the events of armed conflict into the system of impact factors to express the role played by geopolitics in the land-use change of the Indochina Peninsula.

In terms of influencing factors, previous studies have explored the driving mechanisms of PLES from the aspects of physical geography, socio-economics, geo-environment, trade cooperation. The study area generally focuses on hotspots and fragile areas, and there have been few studies of the driving mechanisms of PLES on the Indochina Peninsula as a whole, which is not conducive to land resource planning and integrated regional development [2,10]. Therefore, it was necessary to carry out research on the evolution of the three land uses' spatial patterns and driving mechanisms of the Indochina Peninsula.

The innovation of this paper lies in the introduction of the GTWR model considering the spatial and temporal nonstationarity of the factors, exploring the evolution of the PLES pattern and the driving mechanism of the Indochina Peninsula based on a grid scale, and exploring the land transformation of the region from the perspective of human–land relationship, so as to provide a reference for the subsequent research on the driving mechanism of the land-use change, ecological assessment, and simulation and prediction of the Indochina Peninsula. Previous scholars introduced the GTWR model, and the research scale was mostly based on administrative divisions or meteorological monitoring stations [56,57], which only considered socio-economic factors or natural meteorological conditions, and could not finely express the influencing roles of various factors in the evolution of PLES. In this paper, we considered exploring the influence of four types of factors, namely, humanistic location, social economy, natural environment, and geopolitics, on PLES from the grid scale, which makes the research scale more refined and the factors more comprehensive.

In the actual development process, due to the changes in the complexity of PLES caused by multiple factors, factors such as regional investment level, government policies, and soil properties should be considered in the future to improve the parameters and make the GTWR model fit better.

## 5. Conclusions

This study focused on the evolution of spatial and temporal patterns of PLES on the Indochina Peninsula from 2010 to 2020. It explored the developmental changes in human–land relations, analyzed the driving mechanisms of PLES changes, considered the spatial and temporal nonstationarity of the driving factors, and portrayed the spatial and temporal distributions of, and changes in, the drivers. It, therefore, provides reference information for the land-use function of the Indochina Peninsula and provides new perspectives for the study of driving mechanisms and changes. The conclusions are as follows:

1. The area of interconversion of PLES utilization types in the Indochina Peninsula from 2010 to 2020 was 212,818.70 km$^2$, which was manifested in the conversion of ecological space into productive space and the interconversion of woodland ecological space and grassland ecological space.
2. There was a spatial variation in the rate of change in spatial patterns, with Cambodia having the fastest rate of change in PLES, followed by Laos and Myanmar the slowest.
3. The migration path of the center of gravity of PLES on the Indochina Peninsula demonstrates significant directional differences. In 2010–2020, production space migrated to the southwest, living space shifted to the northeast, and ecological space shifted to the east.
4. The transfer of PLES functional types throughout the Indochina Peninsula was influenced by social context and regional environment, the degree of influence of each factor having significant spatial and temporal heterogeneities. The distribution areas of positive and negative feedback effects for each factor were different, as were the transfer directions.

**Author Contributions:** Conceptualization, C.C.; methodology, X.L.; software, S.L.; validation, S.L.; formal analysis, S.L.; investigation, Q.G.; resources, S.L. and Z.Z.; data curation, Z.Z. and L.Y.; writing—original draft preparation, S.L.; writing—review and editing, C.C. and X.L.; visualization, S.L.; supervision, L.Y. and Z.B.; project administration, C.C.; funding acquisition, M.H. and Z.Z. All authors have read and agreed to the published version of the manuscript.

**Funding:** This research was funded by "Humanities and Social Sciences Project funded by the Ministry of Education", grant number "20YJCZH087", "National Natural Science Foundation", grant number "42202280", "Basic Scientific Research Funds of China University of Mining and Technology (Beijing)—Top Innovative Talents Cultivation Fund for Doctoral Postgraduates", grant number "BBJ2023020", and "Special Fund for Basic Scientific Research Funds of Central Universities and University Student Innovation Training Project of China University of Mining and Technology (Beijing)", grant number "202202051".

**Data Availability Statement:** All data included in this study are available upon request by contacting the corresponding author.

**Conflicts of Interest:** The authors declare no conflict of interest.

## Appendix A

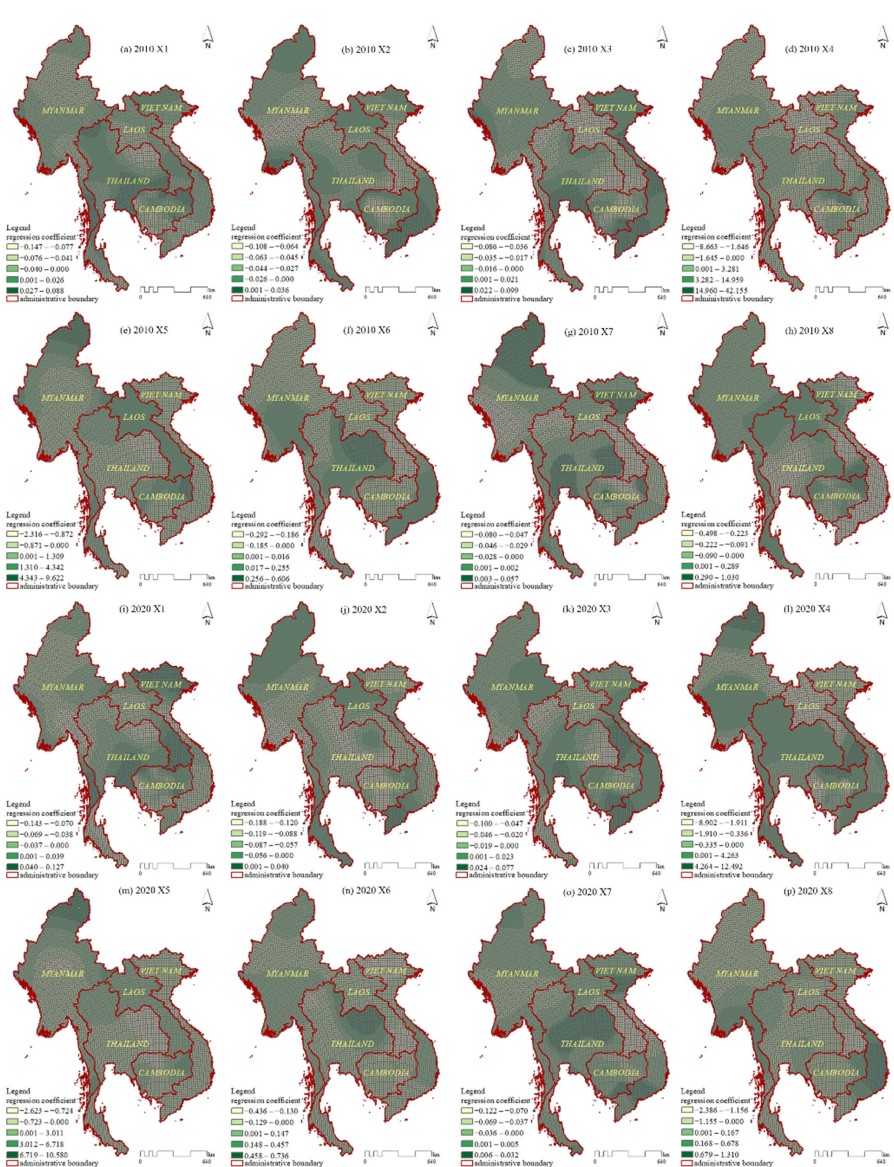

**Figure A1.** Spatiotemporal distribution of the effects of factors on industrial production space.

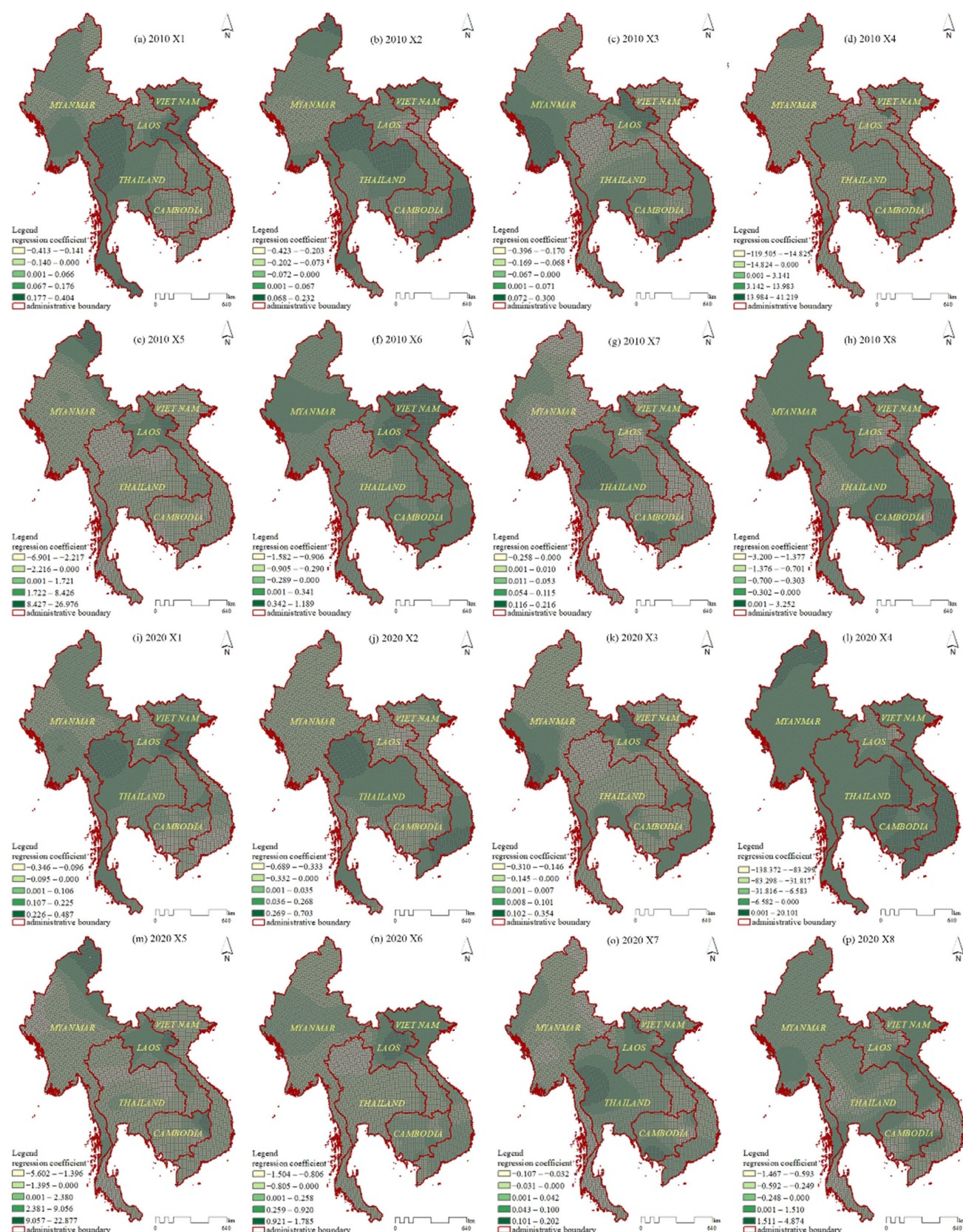

**Figure A2.** Spatiotemporal distribution of the effects of factors on grassland ecological space.

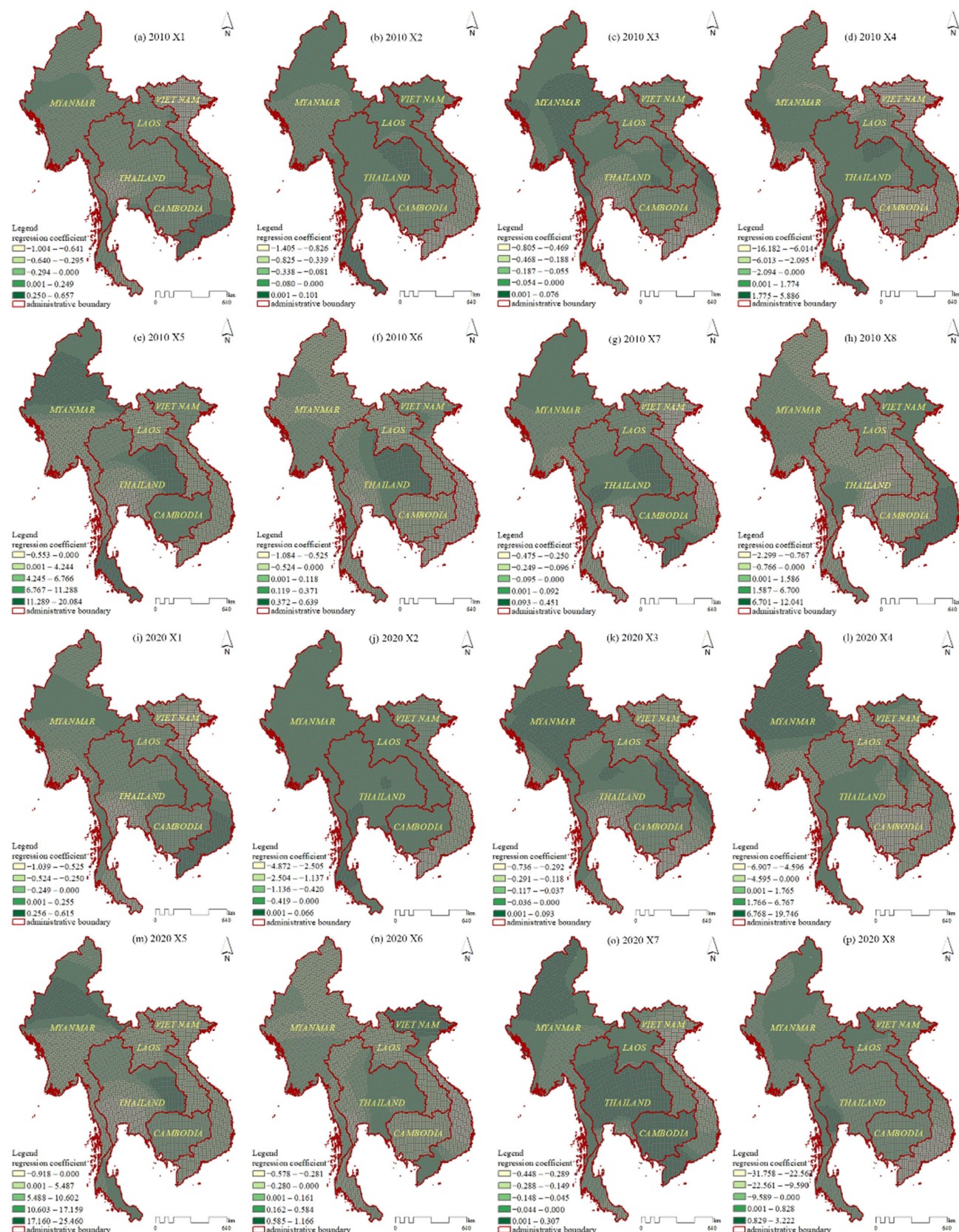

**Figure A3.** Spatiotemporal distribution of the effects of factors on rural living space.

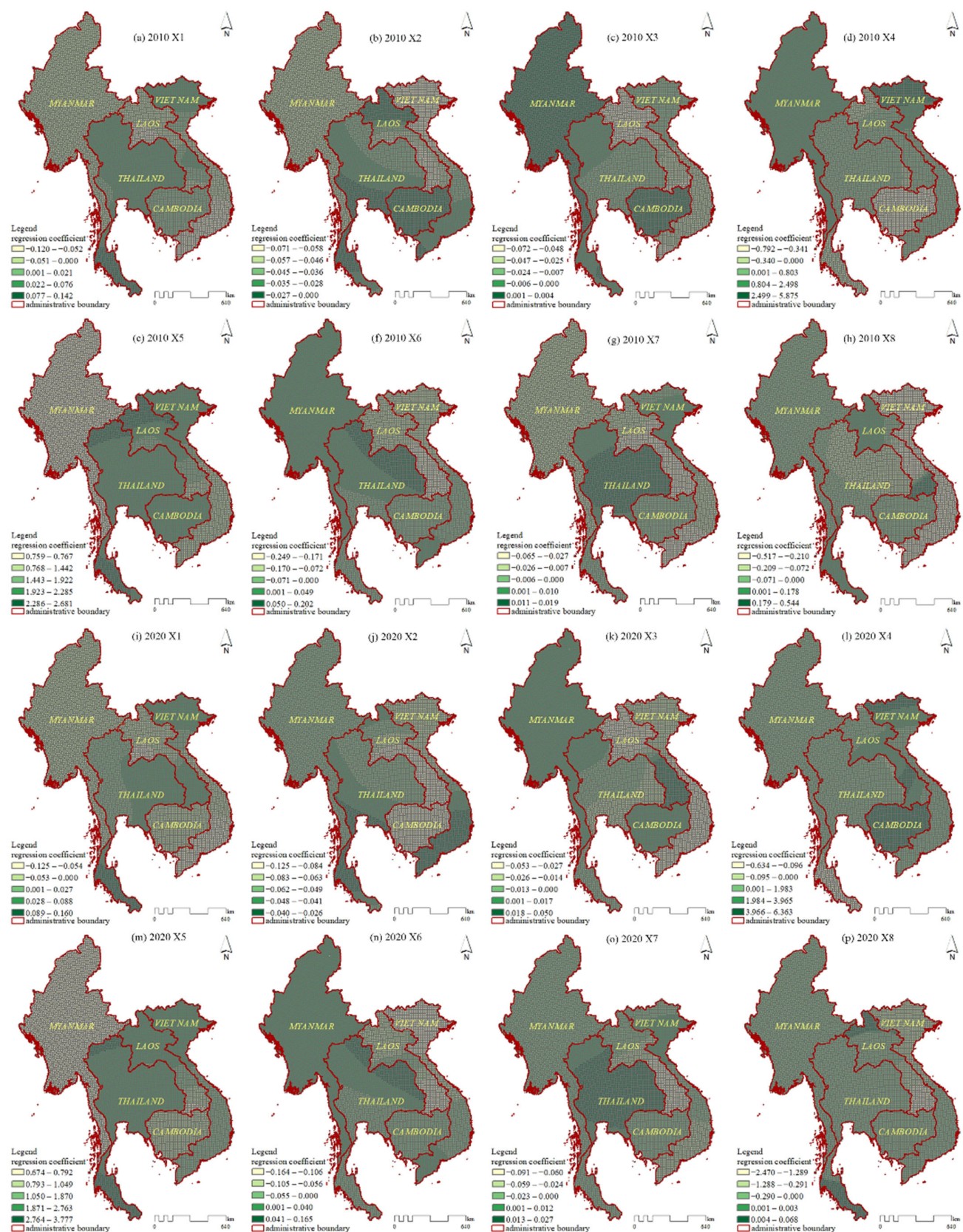

**Figure A4.** Spatiotemporal distribution of the effects of factors on urban living space.

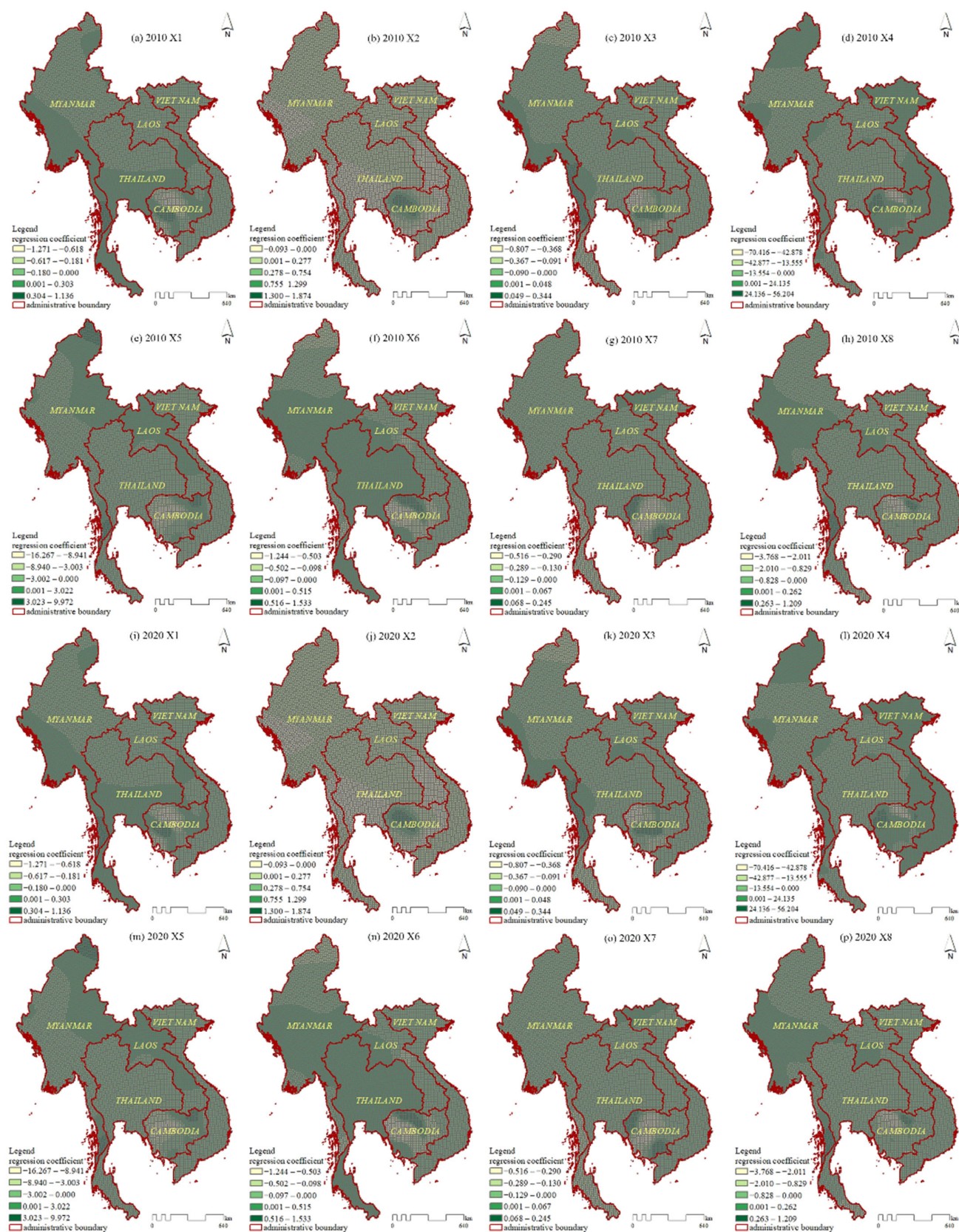

**Figure A5.** Spatiotemporal distribution of the effects of factors on water ecological space.

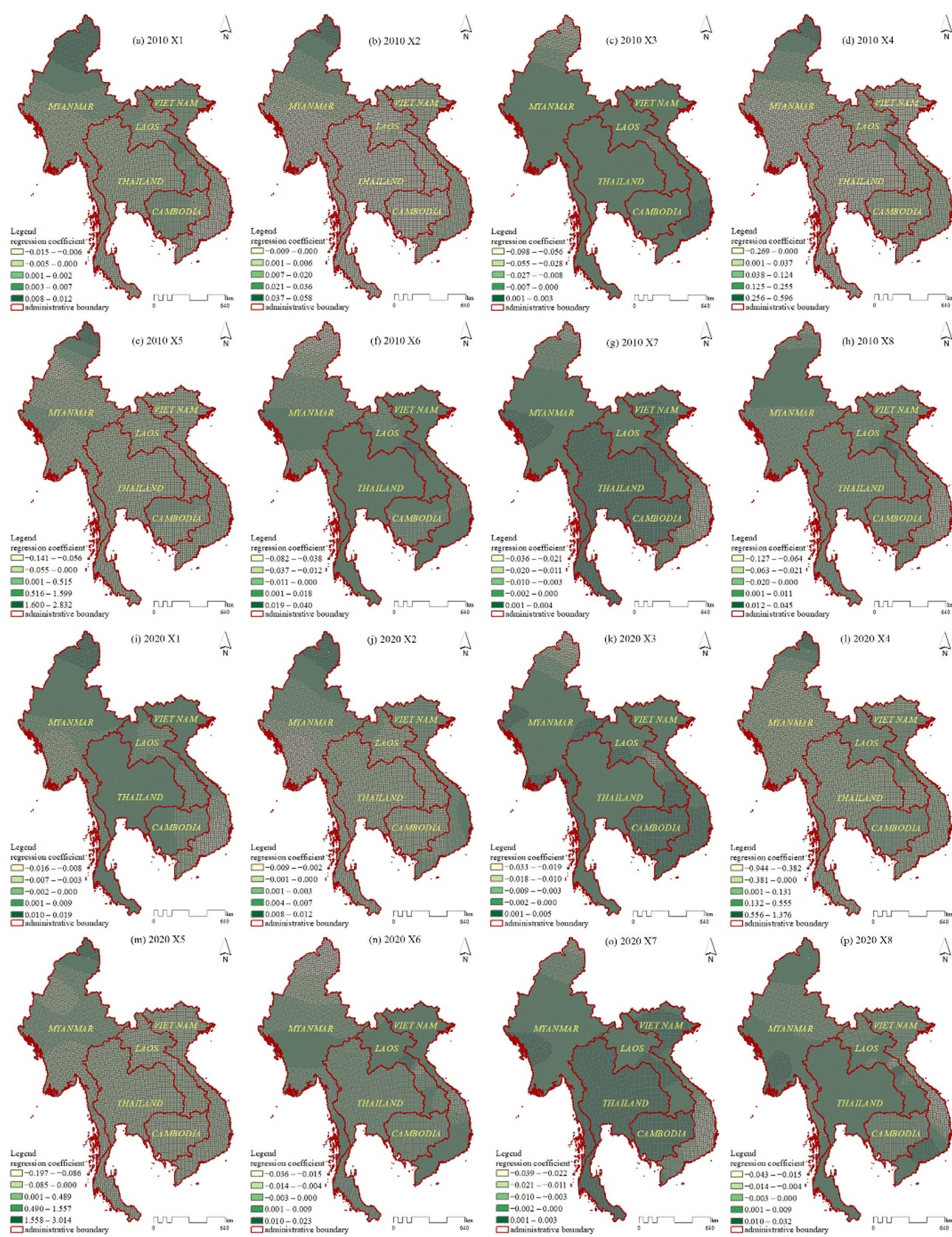

**Figure A6.** Spatiotemporal distribution of the effects of factors on other ecological space.

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
