# Peer review of "Study into the Evolution of Spatiotemporal Characteristics and Driving Mechanisms of Production–Living–Ecological Spaces on the Indochina Peninsula"

_land, doi:10.3390/land12091767_

Round 1
Reviewer 1 Report
The paper/article should clearly focus on the issue; right now the long introduction and the description of the methodology diverts from the complex content and the results. Sometimes it is better to introduce into the methods after the topic and the subject is clearly described...
Reviewer 2 Report
In the article titled "Study into the evolution of spatio-temporal characteristics and driving mechanisms of production–living–ecological spaces on the Indochina Peninsula" and numbered land-2563588; The study focused on examining spatial and temporal changes in production-living-ecological areas (PLES) in the Indochina Peninsula from 2010 to 2020. The article results provide new perspectives in analyzing changes and providing reference information in the land use function of the Indochina Peninsula. In general, the handling and layout of the article is very well prepared. It has a high potential to contribute to the literature in terms of study results. Some minor corrections are recommended.
In the data in “Table 3. Collinearity diagnostics”, two digits are taken in some numbers. All data should be written as standard
The maps in Figure 6 and Figure 7 are of low visibility. Legends are not visible.
Reviewer 3 Report
Some specific comments as follows:
1) The explanation of PLES spatial changes in the abstract should be condensed and the results obtained from the standard deviation ellipse should be elaborated to better explain the evolution characteristics of PLES in the research area. Please give us the goal of this paper in the abstract section.
2) The introduction hardly mentions the research progress in PLES both domestically and internationally, which can appropriately explain the research work of some scholars in PLES. And add the specific research purpose of this article at the end of Introduction section, so that readers can better understand the significance of this article's work.
3) Due to the diversity of PLES classification methods, please add relevant literature on PLES classification in Table 1 as the classification basis.
4) The fishing network in this article is 15km × 15km, have multiple fishing network effects been compared with each other? If so, please explain.
5) I suggest adding diagrams of changes in various spatial types before Figure 3 to provide readers with a more direct understanding.
6) Figure 5 can focus on the position of the center of gravity and the standard deviation ellipse, and display the corresponding spatial types of layers in the map, making readers feel more intuitive. By the way, the figure quality need improvement.
7) I suggest changing “Discussion” into “Conclusion”. Please concise the conclusion section.
8) I suggest adding “Discussion” section to explain the similarities and differences between this article and other PLES related research results, as well as the advantages and disadvantages of the method.
9) A great deal of research work has already been done in this research direction. So, could you tell us your paper novelty?
Round 2
Reviewer 3 Report
The author has revised it carefully. But it still has some minor issues. Please revise it again.
1) The introduction mainly discusses the national research context. The results and conclusions are also only considered in a national context. Therefore, it is recommended to include recent international references in the literature discussion. The international relevance of your findings should be analyzed and provided. 2) At the end of the introduction, a summary of the research objectives is provided to provide a clear overview of the purpose of this study. Please concise the conclusion section, merge the same content. This method helps readers better understand the meaning of this work.
3) The discussion leans more towards the description of the driving mechanism of PLES, and some existing research results of PLES can be added. Firstly, the similarities and differences between this study and other domestic and foreign PLES distributions can be explained, and then the driving mechanism that leads to these similarities and differences can be discussed. By the way, I suggest the author write discussion first, then to write conclusion section.
